# Wrangling Actin Assemblies: Actin Ring Dynamics during Cell Wound Repair

**DOI:** 10.3390/cells11182777

**Published:** 2022-09-06

**Authors:** Justin Hui, Viktor Stjepić, Mitsutoshi Nakamura, Susan M. Parkhurst

**Affiliations:** Basic Sciences Division, Fred Hutchinson Cancer Center, Seattle, WA 98109, USA

**Keywords:** wound repair, actin, cytoskeleton, actomyosin ring, membrane plug, cell cortex remodeling, myosin, Rho GTPases, formins, Wiskott–Aldrich Syndrome family

## Abstract

To cope with continuous physiological and environmental stresses, cells of all sizes require an effective wound repair process to seal breaches to their cortex. Once a wound is recognized, the cell must rapidly plug the injury site, reorganize the cytoskeleton and the membrane to pull the wound closed, and finally remodel the cortex to return to homeostasis. Complementary studies using various model organisms have demonstrated the importance and complexity behind the formation and translocation of an actin ring at the wound periphery during the repair process. Proteins such as actin nucleators, actin bundling factors, actin-plasma membrane anchors, and disassembly factors are needed to regulate actin ring dynamics spatially and temporally. Notably, Rho family GTPases have been implicated throughout the repair process, whereas other proteins are required during specific phases. Interestingly, although different models share a similar set of recruited proteins, the way in which they use them to pull the wound closed can differ. Here, we describe what is currently known about the formation, translocation, and remodeling of the actin ring during the cell wound repair process in model organisms, as well as the overall impact of cell wound repair on daily events and its importance to our understanding of certain diseases and the development of therapeutic delivery modalities.

## 1. Introduction

The cell cortex acts as a physical barrier protecting a cell’s cytoplasmic contents from the external environment. The plasma membrane can easily become compromised as it is under constant attack by a wide variety of external factors that include pore-forming toxins, physical constraints when migrating through a dense matrix, cell shape changes/contractions, or the ravages of diseases; all of which may lead to membrane tears. To combat these stresses, cells of all sizes have developed robust wound repair mechanisms to alleviate the chances of infection or death [1,2,3,4,5,6,7,8]. Cell wound repair is highly conserved and a variety of model organisms and systems have been developed to study this process, including the *Drosophila* syncytial embryo [9], *Xenopus* oocytes [10,11], *Dictyostelium* [12], sea urchin eggs [13], *C. elegans* [14], and tissue culture cells [15,16]. These have proven to be excellent and complementary systems for studying this dynamic cell biological process and for identifying the molecular players/pathways involved. Uncovering the cellular and molecular mechanisms underlying cell wound repair may aid in our understanding of critical cell behaviors and fundamental biological regulations, as well as in pathological states from infections and diseases/cancers to the development of regenerative medicine therapies.

While the molecular details can vary among the different cell wound repair models, the physiological/cell biological sequences of events to close a wound following a breach to the cell cortex are similar, especially among larger sized wounds (>100 nm in diameter) (Figure 1A). The first detectable event following disruption of the cell cortex that sets the repair process in motion is a sudden influx of extracellular calcium (Figure 1B) [10,17,18,19]. In response, the cell rapidly reseals the plasma membrane to prevent excessive loss of intracellular content. To achieve this, vesicles are recruited to the injury site and form a temporary membrane patch to plug the hole [18,20]. While the mechanisms of wound closure after rapid resealing of the wound hole are context-dependent [2,5,8], in the larger cell *Xenopus* oocyte and *Drosophila* embryo models, an F-actin ring is organized around the injury site and is then translocated inward to close the wound (Figure 1C) [1,2]. Once the wound fully closes, the temporary membrane patch is removed from the wound site by internalization and/or extracellular release (Figure 1A). These steps are followed by a remodeling phase wherein the actin ring is disassembled, and the plasma membrane and underlying cytoskeleton are refashioned to their pre-wounded state (Figure 1D).

Similar to the *Xenopus* oocyte model, wounds generated within a smaller single cell of the *Xenopus* embryo epithelia exhibit accumulation of F-actin [22] and Rho family GTPase reporters at the wound periphery [21] (Figure 1E).

To visualize repair dynamics, a variety of fluorescent reporters or dyes are used in different cell wound systems. For example, since the actin cytoskeleton aggregates in the interior of the wound, rather than forming a ring that constricts around the wound periphery, membrane dyes such as FM4-64 are widely used to monitor membrane resealing in tissue culture cell models [23,24]. As the cortical cytoskeleton is highly linked to the overlying plasma membrane and an actin ring is formed at the periphery of wounds in the *Xenopus* and *Drosophila* models, membrane dyes and actin reporters can be used interchangeably to monitor repair dynamics in these models (see Figure 2). Membrane dyes can mask the wound edge due to their labeling of the temporary membrane patch, as well as being endocytosed into cells during live imaging. Fluorescent F-actin reporters are therefore widely used to monitor dynamic wound closure over time in larger wound models.

As expected, this dynamic repair process requires the intricate orchestration of many proteins and protein families to perform its specific functions, including Rho family GTPases, actin nucleators, myosin, annexins, and actin cross-linkers [17,24,25,26,27]. Interestingly, a number of these key components are dysregulated in human pathologies that exhibit aberrant wound repair, including Duchenne muscular dystrophy [28], Miyoshi myopathy [29], Diabetes [30], and invasive bladder cancer [31]. Thus, single cell wound repair models are vital tools for studying pathology-associated proteins and their interaction partners in an inducible and dynamic system. In this review, we focus on the orchestration of actin and actin binding proteins to form, translocate, and remodel the actin ring through the single cell wound repair process in larger cell wound models.

## 2. Actin Ring Formation

Following the rapid plugging of the wound, a cell must assemble a Rho family GTPases-regulated actin ring at the wound periphery to pull closed the cell cortex [21,26]. This is accomplished in three main steps: actin stabilization, actin recruitment to the wound, and actin organization at the wound periphery [1,2].

### 2.1. Actin Stabilization at Wounds

The Annexin family proteins are membrane-bound calcium-dependent proteins that are crucial for the organization and regulation of the plasma membrane. Annexins also serve as an important bridge between the plasma membrane and the cortical cytoskeleton [32]. During cytokinesis, actin filaments are bundled and stabilized against the plasma membrane, and mammalian Annexin A2-depleted cells show defects in cytokinesis [33,34]. Mammalian Annexins are also required for proper cell wound repair, facilitating the spatial localization of proteins such as dysferlin in muscle cells [24,35,36,37,38,39,40,41]. In the *Drosophila* model, Annexin B9 is rapidly (<3 s) recruited to wounds to stabilize actin. This actin stabilization is necessary to allow the recruitment and formation of a RhoGEF2 array around the wound site [27]. Two other *Drosophila* Rho GEFs (RhoGEF3 and Pebble) are also recruited to the wound periphery, where they are arranged in a pattern of concentric rings [27] (Figure 3A).

### 2.2. Actin Recruitment to Wounds

The pre-pattern formed by RhoGEFs at the wound directs the subsequent recruitment of Rho family GTPases (Rho, Rac, Cdc42) into similar concentric arrays at the wound periphery (Figure 3B–D) [21,26,27]. Rho family GTPases are known to regulate the rapid recruitment of actin (“cortical flow”) to wounds where it can undergo polymerization, bundling, and crosslinking to form the dense actin ring. In addition to actin recruitment, such cortical flow of actin is also observed in other cellular processes, such as cell division and cell migration, and contributes to the generation of force, as well as the organization/compaction of actin filaments [42,43,44].

In the *Drosophila* cell wound model, Rho is necessary for actomyosin ring organization and stabilization, whereas Rac, and Cdc42 to a lesser extent, are necessary for actin mobilization towards the wound (Figure 3E) [26]. Knockdown of Cdc42 was shown to result in less-oriented actin cortical flow towards the wound, whereas knockdown of Rac severely disrupted the directed actin cortical flow [26]. In the *Xenopus* model, the correct concentric circle zoning of Rho GTPases is dependent on the relative amount of each Rho GTPase present (i.e., the Cdc42 zone width is dependent on the abundance of Rho and vice versa). Minor changes to this Rho GTPase patterning can affect the rate of wound closure [45].

### 2.3. Actin Organization at Wounds: Rho Family GTPases

Once actin is recruited to the leading edge of the wound, it is organized into different functional assemblies that must be precisely and dynamically coordinated to allow the efficient translocation of the actomyosin ring. Rho family GTPases, through their downstream effector proteins, organize the recruited actin such that it forms a dense ring bordering the wound edge (see Figure 1C) [10,26,27]. Downstream effectors of Rho family GTPases include linear and branched actin nucleation factors—both of which are important for the generation of distinct actin ring architectures.

#### 2.3.1. Linear Actin

Linear filaments, required for normal cellular processes including cytokinesis and filopodia formation, are regulated by *de novo* nucleation promoting factors and bundling proteins, which govern filament formation rates and organization, respectively (Figure 4A,B) [46,47,48]. Linear actin nucleation factors function downstream of Rho and include Diaphanous-related formins (DRFs) that directly nucleate linear actin filament assembly *de novo* and, whose knockdown results in disrupted actin ring formation during cell wound repair (Figure 4C,D) [26]. Formins elongate filaments until they dissociate from the parent filament and allow capping protein to terminate elongation [49,50]. Simultaneously, linear actin filaments are bundled by crosslinkers to form robust actin structures. For example, α-actinin links antiparallel actin filaments to make larger and more ridged bundled filaments that possess dynamic bundling patterns crucial for cytokinesis in fission yeast (Figure 4E) [51].

#### 2.3.2. Branched Actin

Branched actin nucleation promoting factors include the Wiskott–Aldrich Syndrome (WAS) family of proteins: WASp, SCAR/WAVE, and WASH (cf. [52]). WAS family proteins polymerize branched actin through their interaction with the Arp2/3 complex (Figure 4B,F) [52,53,54,55,56,57,58,59]. Branched actin filaments, along with linear filaments, are necessary for the assembly of different higher order architectures. The presence of ordered branched actin structures is crucial for actin ring formation, as well as its subsequent contraction to effectively close a wound. Indeed, in the absence of branched and linear nucleation promotion factors, no actin structures form [60,61]. In general, WAS family proteins have a one-to-one correspondence with Rho family GTPases (Cdc42 > WASp; Rac > SCAR/WAVE; Rho1 > Wash), where each pair regulates specific dynamic actin organizations. However, WAS proteins have recently been shown to have non-redundant roles during cell wound repair. In particular, WASp has a specialized function to regulate the orientation of actin filaments making up the contractile actomyosin ring at wound periphery [61].

### 2.4. Actin Organization at Wounds: Other Factors

While Rho family GTPases and their downstream effectors are indispensable for the recruitment and organization of actin at wounds, some actin organization is still present upon wounding in the absence of Rho family GTPases, suggesting that other factors must be involved. One such factor, Pavarotti (Pav), is a kinesin-like microtubule-dependent molecular motor protein that regulates cytoskeleton dynamics during cytokinesis, neuronal migration, and neurite outgrowth [62,63,64,65,66,67]. In particular, Pav makes a protein complex with a RhoGEF and a RacGAP during cytokinesis, which then activates Rho1 to form a contractile actomyosin ring at the cleavage furrow. Pav, while usually associated with microtubules, has recently been shown to have a non-canonical role in regulating actin dynamics through directly binding to actin and bundling it [68]. This actin association of Pav is needed for its role in cell wound repair: Pav knockdown does not affect the recruitment of Rho family GTPases to wounds, but leads to delayed actin recruitment to wounds, weak actomyosin ring formation, and a delayed wound closure rate [68]. Pav’s actin bundling activity contributes to the robustness of the actin ring by aligning the actin filaments and/or stabilizing the established actin ring at the wounds.

In addition to Pav, knockdowns of some canonical insulin signaling components have been shown to affect actin dynamics during cell wound repair using the *Drosophila* model [68]. While the involvement of insulin signaling in cell wound repair was thought to be a secondary effect (e.g., due to the weakening of the plasma membrane [30]), insulin signaling is activated upon wounding where it regulates the downstream actin regulators, Girdin and Chickadee. Girdin is an actin crosslinker that regulates actin bundling, stabilization, and network formation [69]. Girdin interacts with the catenin–cadherin complex [70] and contributes to the formation of robust actin ring as well as Pav and E-cadherin functions in cell wound repair. Chickadee is the Drosophila homolog of Profilin that binds to monomeric actin. Actin–Profilin complexes associate with different types of actin nucleators (i.e., formins, VASP, and WASP), where it is essential for the F-actin polymerization [71,72].

### 2.5. Actin Architectures in Cell Wound Repair

Actin and myosin can be arranged in a number of different functional assemblies within cells, including actomyosin cables and stress fibers. Within these assemblies, actin filaments and myosin can organize into a wide variety of dynamic structural architectures based on their spatial organization and connections, leading to different proficiencies in generating forces necessary for pulling the wound closed [60,73,74,75,76,77]. Different combinations of actin filament types and orientations have been described, including disordered networks, branched actin meshes, and disordered bundles (Figure 4G) [60,73]. The precise combination and organization of actin can have drastic effects on the efficiency of actin ring contraction [60]. In vitro actomyosin rings consisting of only linear or only branched actin exhibit impaired contraction which can be modulated by the presence of varying concentrations of actin accessory proteins such as α-actinin and cofilin. Thus, an elaborate and dynamic interplay of the linear and branched actin networks and various actin binding proteins is necessary for robust cell wound repair.

## 3. Actin Ring Translocation

After proper assembly of the actomyosin ring, the cell must coordinate its translocation to close the wound and thereby repair the injury site. In comparison to the assembly of the actin ring (~30 s in the *Drosophila* model), translocation of the ring to close the wound is a much slower process (~10 min), during which there must be a continuous and dynamic spatiotemporal recruitment of several key players such as Rho family GTPases [14,25,27,45], the Arp2/3 complex [14,78], and non-muscle myosin II (myosin) [26]. Their roles in actin ring translocation are highlighted by their persistent expression at the wound edge. Other actin binding proteins have also been demonstrated to be vital during actin ring translocation such as the actin-associated proteins α-actinin, cofilin, and anillin [60,79]. These known contributors can affect wound repair by facilitating dynamic actin ring organization and thus the ability for the actin ring to efficiently translocate.

### 3.1. Diversity in Translocation Mechanisms

Although several model organisms share the same groups of recruited proteins, constriction of the assembled actin ring to close the wound can be accomplished through different mechanisms (Figure 5). Several mechanisms have been proposed for actin filament network contractility: (1) sarcomere-like contraction, involving filaments sliding due to motor processivity and crosslinker distribution; (2) actin filament treadmilling, involving continuous actin assembly at the inner edge of the ring and disassembly at the outer ring edge; and (3) F-actin buckling model, involving the mechanical deformation of actin filaments by myosin [9,25,60,61,80,81]. However, under certain conditions, cells deficient in one or more of the cytoskeletal components will still attempt to repair wounds. A recent study has demonstrated that a *Drosophila* embryo lacking the majority of branched actin and myosin exhibits actin filament swirling to close a wound [61]. Further investigations will be required to uncover the molecular players and pathways underlying new mechanisms of wound closure.

#### 3.1.1. Actomyosin Ring Contraction

In the *Drosophila* model, myosin is recruited to the wound edge to form and contract the actomyosin ring [9]. Using pharmacological inhibition, myosin has been demonstrated to be indispensable during actin ring translocation: where actin rings in treated embryos have shown to be unable to close wounds, highlighting the requirement for myosin [26] (Figure 5A,B). Several other components and characteristics of the actomyosin ring can greatly impact the efficiency and effectiveness of myosin-dependent constriction, including actin filament orientation, organization, and connectivity. Using in vitro and in silico assays, the actin crosslinker α-actinin has been demonstrated to modulate the contraction of different actin architectures and facilitate symmetric contraction of actin rings [60,82]. These studies have demonstrated the requirement for a connected filament network in the actin ring to ensure the proper distribution of tensional forces exerted by myosin. Additionally, the F-actin depolymerizing protein, ADF/cofilin, has been suggested to dynamically reorganize actin ring connectivity during contraction [60] and proposed to play a vital role in cytokinetic ring contraction in budding yeast [83,84]. Taken together, these studies demonstrate the importance of actin network connectivity and reorganization to myosin-dependent actin ring contraction during wound repair.

#### 3.1.2. Actin Treadmilling

*Xenopus* oocytes recruit myosin to the wound edge to form and contract the actomyosin ring [85,86]. However, using pharmacological inhibition, myosin has been demonstrated to be dispensable, and the actin ring was able to translocate to close the wound, albeit at a slower rate (Figure 5C,D) [25]. It has been revealed through active RhoA and Cdc42 labeling that there was greater GTPase activity at the leading edge of their recruitment zones and reduced activity at the trailing edge. In this way, actin is continuously assembled at the inside of the actin ring and disassembled at the outer edge—termed actin treadmilling—to decrease the wound area. As the closure rate solely through actin treadmilling is slower than wildtype conditions, actin treadmilling likely complements myosin-dependent contraction. Additional investigations using this system have identified crosstalk among Rho and Cdc42 to regulate each other’s spatial patterning [45]. These observations solidify the hypothesis that continued regulation of each Rho GTPase is also crucial for proper actin ring translocation.

Similar to *Xenopus* oocytes, *Dictyostelium* cells, *C. elegans* hypodermal cells, and sea urchin coelomocytes have demonstrated myosin-independent wound closure mechanisms [12,14,78,87]. Wounded *Dictyostelium* cells did not exhibit additional myosin recruitment to the injury site and myosin null cells were able to repair wounds comparable to wild-type cells [12,87]. Likewise, coelomocytes treated with the kinase inhibitor KT5926 did not affect wound closure, indicating a myosin-independent wound closure mechanism [78].

#### 3.1.3. F-Actin Buckling Model

The F-actin buckling model incorporates a phenomenon where a single filament exhibits compressive or tensile forces on different portions of the filament. This occurs when numerous myosin filaments are randomly dispersed along an actin filament causing asymmetric forces to be applied along the actin filament [80]. In these scenarios, portions of the filament under compressive forces are more likely to buckle/bend, thus shortening the effective length of the filament (Figure 5E,F) [80,81]. Considerable effort has been made to investigate F-actin buckling using a minimal actin cortex [88], ghost cells [89], and micropatterned surfaces [60]. These systems have demonstrated the role of F-actin buckling to break filaments to provide space for myosin to cluster F-actin and form foci. This is reminiscent of the latter end of actin ring constriction where actin ring-associated filaments shrink and compress to an actin-dense region. This mechanism has been proposed to be pivotal for highly connected actin rings where F-actin buckling contributes to actin turnover and facilitates overall actin ring rigidity [60]. While F-actin buckling has not been observed in any cell wound repair model to date, F-actin buckling may work in concert with actin disassembly proteins to reduce network connectivity and allow actin rings to constrict and pull the membrane closed during cellular wound repair. Alternatively, force generation by shortening F-actin filaments without forming the actin ring may contribute to wound closure in other cell wound models that do not require an actomyosin ring.

### 3.2. Plasma Membrane–Cortical Cytoskeleton Attachment

Since the cortical cytoskeleton and plasma membrane rely on one another for structural support and tension, a vital aspect of actin ring formation and translocation during wound closure is its ability to maintain its connection to the overlying plasma membrane, to simultaneously draw the membrane and cortical cytoskeleton closed. Mathematical modeling has demonstrated the importance of plasma membrane–cytoskeleton anchors for tension generation in the contractile actomyosin ring (Figure 6) [90,91]. These models highlight the role of anchors to both attach filament ends to the membrane and to provide resistance against myosin pulling to generate and accumulate tension [90]. Several proteins and protein families are known to bridge the connection between the cortical cytoskeleton and the plasma membrane. To date, DE-Cadherin and Annexins have been shown to be vital transmembrane and membrane-binding proteins, respectively, during single cell wound repair [9,27,32,35,38,92]. Cadherins are a class of cell–cell adhesion proteins that conventionally play a role in cell–cell binding and tissue morphogenesis, where they link to the cortical cytoskeleton through their associations with the catenin complex [93]. Interestingly, E-cadherin has been shown to accumulate at the wound edge during cell wound repair. Knockdown of E-cadherin in *Drosophila* results in wound overexpansion phenotypes, along with slower wound closure [9]. Similarly, Annexins are required for actin filament stabilization [27,33,34]. With these connections established, linear and branched filaments can then be bundled, modified, and organized into unique actin architectures that are conducive to a stable, yet dynamic, actin ring.

## 4. Cell Cortex Remodeling

Once the wound is fully closed, the cell cortex must undergo extensive remodeling to remove repair structures (i.e., membrane plug and actomyosin ring remnants), and to restore the original cell cortex organization, connections, and functions. These remodeling events are similar to those used during normal developmental events, including the disassembly/removal of spindle remnants following cytokinesis, and the rapid cell cortex reorganizations that allow the dynamic extension and retraction of cellular protrusions during cell migration/metastasis. Despite the conserved requirement for cortex remodeling in response to injury across phyla, less is known about the molecules, machineries, and pathways regulating the remodeling process in this context. Clues to the molecular mechanisms underpinning the remodeling process are beginning to emerge from recent global studies using microarray and RNA-seq analyses in the *Drosophila* embryo and human MCF7 cancer cell models [19,94].

### 4.1. Microarray Studies in the Drosophila Cell Wound Repair Model

Cell cortex remodeling is a significant part of the repair process in the *Drosophila* model: while the wound is fully closed within ~10 min, the subsequent remodeling of the plasma membrane and cortical cytoskeleton takes on the order of 15–30 min. When performing microarray analyses to identify genes whose transcription was up- or down- regulated during the latter half of the *Drosophila* cell wound repair process, several genes were identified that, when knocked down, led to defects in cell cortex remodeling: premature actomyosin ring disassembly or persistent actomyosin ring presence (Figure 7) [19].

One gene that resulted in premature actomyosin ring disassembly is Nullo, a regulator of actin-myosin and adherens junction stability [19,95,96,97]. In the absence of Nullo, the actomyosin ring does not remain compact during contraction and does not appear to be anchored to the plasma membrane, resulting in an unstable actin ring that is degraded quickly.

In contrast to Nullo, Exu knockdown resulted in persistent actomyosin ring presence. Exu is known to regulate mRNA localization during oogenesis [98,99,100,101,102]. Stored mRNAs are locally translated into proteins, which is important for many normal developmental processes (e.g., RhoA is locally translated in developing axons and growth cones [103]). Since inhibition of transcription and translation impairs repair processes [19], mRNAs regulated Exu might be locally translated at wounds to regulate actin dynamics.

### 4.2. RNAseq Studies in the Wounded MCF7 Cancer Cell Model

A global RNA-seq screen using detergent-induced wounds in MCF7 cells showed that MAPK signaling (including p38 and ERK) and pro-inflammatory/immune response molecules are up-regulated just after plasma membrane resealing [94]. While those molecules were not tested directly in cell wound repair, several of them are known to be involved in the regulation of actin dynamics in different cell types and cellular processes. p38 signaling is involved in the reorganization of F-actin and the regulation of RhoA and Cdc42 activity, whereas ERK1/2 associates directly with cofilin and regulates actin disassembly [104,105,106]. Actin also needs to be reorganized for proper plasma membrane remodeling in some cell wound repair models [107]. Once the wound area is resealed, actin-rich protrusions are generated at wound sites to form plasma membrane ruffles. Subsequently, those ruffles become a cup-like structure that is internalized by macropinocytosis to remove the damaged membrane.

While it is clear from these global screens that a number of genes are likely playing a role in the cortex remodeling process based on their temporal timing and gene functions, further studies are needed to reveal the molecular mechanisms involved in restoring cell cortex organization and functions following wound closure.

## 5. Relationship to Diseases and Infections

Robust wound repair mechanisms are essential in living organisms of all shapes and sizes since injuries to individual cells occur frequently in response to daily wear-and-tear, accidents, trauma, violence, clinical interventions, and pathological conditions ranging from infections to diseases and cancers [1,4,5,8,108]. Such wounds are of particular concern when occurring in a non-renewing, irreplaceable cell type, or when in combination with fragile cell disease states such as diabetes, skin blistering disorders, atopic dermatitis (eczema), cardiopathies, and muscular dystrophies. Imbalances of cellular components, either through overproduction or lack of production, can throw off the healing processes. In addition, physiological conditions such as fibrosis, inflammation, and tumorigenesis can be triggered or aggravated by abnormal and/or excessive repair [109,110,111,112].

Skeletal muscles are under constant mechanical stress and are highly subject to damage, making effective cell wound repair crucial for their cell viability. Different muscular dystrophies are associated with defective cell wound repair. For example, mutations in muscle specific genes such as the calcium sensing sarcolemma protein Dysferlin and the endocytosis required protein caveolin-3, which lead to different human limb girdle muscular dystrophies (LGMD2B and LGMD1C, respectively), are required for cellular wound healing [40,113,114,115,116,117,118]. Skeletal muscle myopathies are a common complication in individuals suffering from diabetes and result from aberrant cellular wound repair in this tissue [29,119,120].

Diabetes mellitus is a disease that burdens over four hundred million people in the world [121,122,123]. High glucose levels lead to a myriad of symptoms, including vascular stiffness leading to poor circulation and impaired epithelial repair. Mouse C2C12 myoblast cells cultured in a high glucose environment showed poor membrane repair after just eight weeks of exposure [30]. Interestingly, the canonical insulin signaling pathway is a central component of the cell wound repair process, where it controls actin dynamics through the actin regulators Girdin and Chickadee (profilin). In particular, RNAi knockdown of several insulin signaling pathway components in the Drosophila model resulted in abnormal wound repair with diminished fragile actin rings and delayed wound closure [19]. These findings are consistent with studies showing that the insulin signaling pathway controls aspects of actin dynamics during cell migrations by regulating profilin expression [71].

Many types of malignant cancers have hyperactive wound repair pathways. For example, for cancer cell migration to occur, cancer cells must often be able to sustain considerable physical stress as they squeeze through various types of tissues. Molecules required for repair to occur under these circumstances are emerging. One such protein family, Annexins, are fast-responding calcium-regulated proteins that mediate cytoskeleton stabilization and cell cortex remodeling, and whose expression is often upregulated in response to injury [27,32,33,35,38]. Different Annexins have been shown to affect many aspects of the cell wound repair process, ranging from their effects on actin architecture and organization to the modulation of cortical tension associated with bringing wound edges together [27,36,37,39,124,125,126].

Assault by microbial pathogens (bacterial, viral, fungal) is another serious threat to cell viability: pathogen invasion damages the cell cortex and triggers the cell wound repair response [2,4,5,8]. Many of these wounds lead to small breaches in the cell cortex that are removed by endocytosis of the membrane region including the lesion (cf. [127]) or by membrane shedding/cytosolic purging to remove large portions of damaged membrane [128,129,130,131,132]. Some wounds such as chronic ulcers, can become colonized by antibiotic resistant bacteria that release pore-forming toxins (PFTs). These toxins can oligomerize and insert themselves into the membrane, creating a cavity that disrupts osmotic balance and cell viability, and triggers a more substantial wound-healing response [133,134]. While cytoskeletal remodeling and membrane changes have been observed in the context of PFT lesions, the mechanisms of actin dynamics surrounding such repairs are just beginning to emerge [128,134,135,136]. Thus, delineating the molecular basis of cell wound repair is of profound clinical relevance, both for understanding disease pathologies and infections, and for designing effective treatments/therapies.

## 6. Conclusions and Future Directions

Cell wound repair consists of three major steps: membrane resealing, formation then contraction of an actin ring, and cortex remodeling to restore the original organization/function. Here we have focused on the roles of actin and actin binding proteins on actin ring formation, translocation, and remodeling during the repair process. Formation of a robust actin ring requires highly coordinated functions of Rho family GTPases, linear/branched nucleators, and crosslinkers to recruit, compact, and orient actin filaments around wounds. While molecules are very conserved among different models, the molecular make-up of ring structures are context-dependent (myosin-dependent and -independent rings). It is still unclear what the exact actin ring architectures are in the different repair models or how actin dynamics are regulated to form them for efficient wound closure.

After an actomyosin ring is formed, several context-dependent mechanisms have been described for actin ring translocation: actomyosin contraction, actin treadmilling, and F-actin buckling. While these mechanisms explain how the actin ring can translocate, little is yet known about what determines which translocation method will be used in each context. When wounds fully close, the cell cortex must be remodeled to restore the original function. It is not known how cells know the wound has completely closed or if/when/how to initiate the cortex remodeling step. Phenotypes of pre-mature actin ring disassembly and pro-longed actin accumulation in the *Drosophila* model suggest that control of the timing of remodeling might require specific pathways rather than just being able to detect hole closure and physical cell membrane adhesion alone. Recent transcriptome analyses during cell wound repair have provided a good set of candidate genes to start investigating the genes/pathways involved in actin ring remodeling.

The precise spatial and temporal control of protein recruitment around wounds following calcium influx is key for all subsequent steps of wound repair. In addition to investigating each step of the repair process, many questions remain concerning the transitions between steps. There are still significant gaps in our understanding of the mechanistic events governing these different steps: between calcium influx and recruitment patterns to the wound edge of responding proteins such as those of Rho family GTPases, as well as between actin ring formation and the initiation of ring translocation to pull the wound close. In combination with super-resolution microscopy, recent advances in optogenetic techniques have provided exciting new avenues and opportunities for addressing these open areas.

The study of cell wound repair has considerable clinical relevance. Pharmaceutical delivery methods and technologies are a popular and growing field of research that relies heavily on an understanding of the fundamental mechanisms of local and reversible plasma membrane disruptions to design effective methods of drug delivery [137,138,139]. Understanding the full spectrum of molecular mechanisms underpinning cell wound repair will provide important new insights into the many critical cell behaviors and fundamental biological regulations that take place during cellular events in daily life, and in pathological states from infections to diseases/cancers.

## Figures and Tables

**Figure 1 cells-11-02777-f001:**
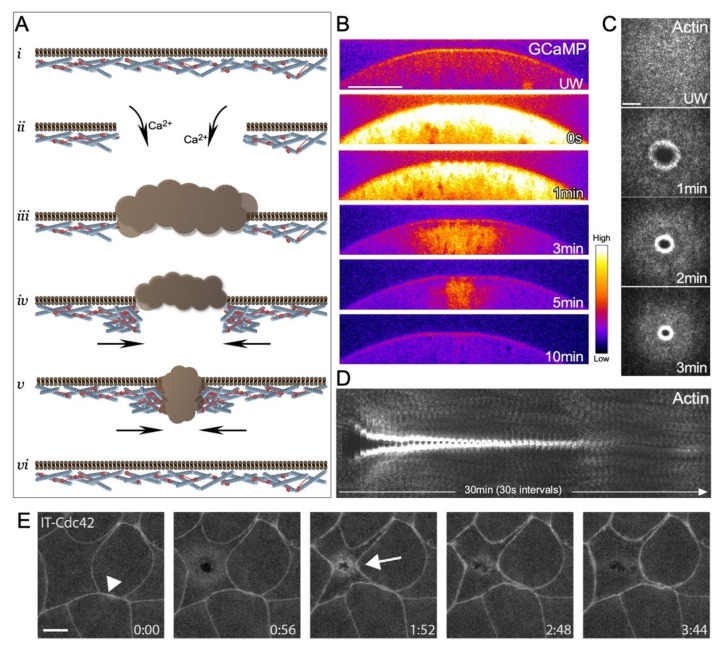
Aspects of single cell wound repair in model organisms. (**A**) Schematic of the major steps in cell wound repair: (*i*) unwounded plasma membrane and actin cytoskeleton; (*ii*) initial influx of calcium ions initiating the wound repair process; (*iii*) temporary vesicular membrane patch formed to quickly plug the injury site; (*iv*) formation of an actomyosin ring in the cortical cytoskeleton around the injury site; (*v*) translocation of the actomyosin ring to pull the wound closed; (*vi*) remodeling of the cell cortex to remove the temporary membrane patch and the actomyosin ring, and to restore homeostasis. (**B**) Confocal micrographs of cross-section of a syncytial *Drosophila* embryo expressing a calcium reporter (GCaMP) during the wound repair process. Scale bar: 20 µm. (**C**) XY maximum confocal micrographs of the actin ring formed post laser injury in a syncytial *Drosophila* embryo. Scale bar: 20 µm. (**D**) XY kymograph across the wound area depicted in (**C**). During the wound repair process, a robust actin ring forms around the wound, translocates inward to reduce the wound area and eventually disassembles when the wound is closed. (**E**) Example of a single cell wound within the context of the epithelia of a *Xenopus* embryo showing the accumulation of a Cdc42 reporter at the wound periphery (arrow) and at cell-cell junctions (arrowhead). Scale bar: 10 µm. Reprinted from Golding et al., (2019), eLife 8: doi:10.7554/eLife.50471 [21].

**Figure 2 cells-11-02777-f002:**
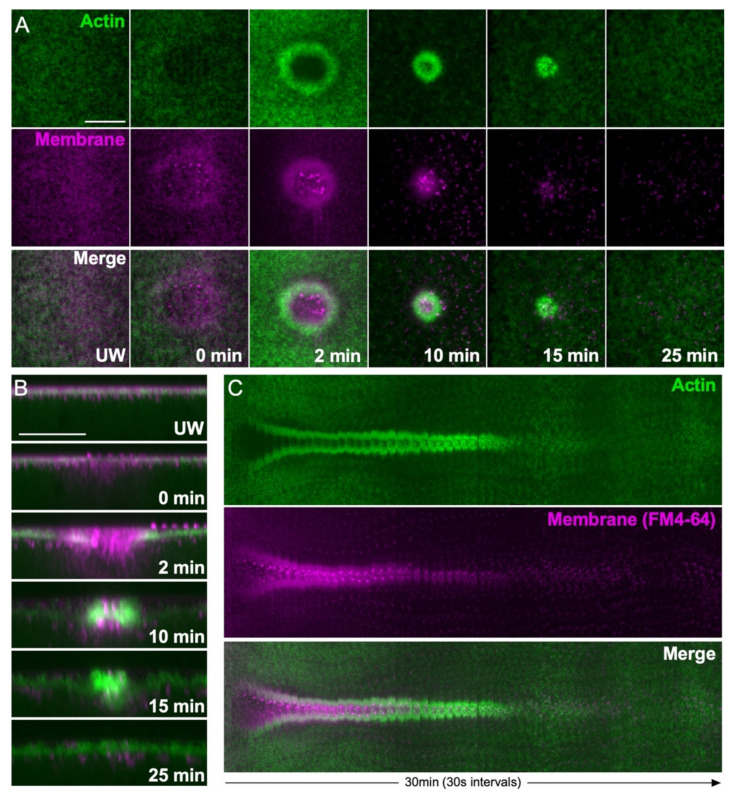
Actin ring and membrane dynamics during cell wound repair in the *Drosophila* model. (**A**) XY projection confocal micrographs of a syncytial *Drosophila* embryo expressing an actin reporter (green) and injected with a membrane dye (FM4-64, magenta) during the wound repair process. Scale bar: 20 µm. (**B**) Confocal micrographs of cross-sections from the images depicted in (**A**). Scale bar: 20 µm. (**C**) XY kymograph across the wound area depicted in (**A**). During the wound repair process, a temporary membrane patch reseals the hole, then an actin ring at the wound periphery pulls the plasma membrane closed. The temporary membrane patch is eventually removed from the wound site by internalization and extracellular release.

**Figure 3 cells-11-02777-f003:**
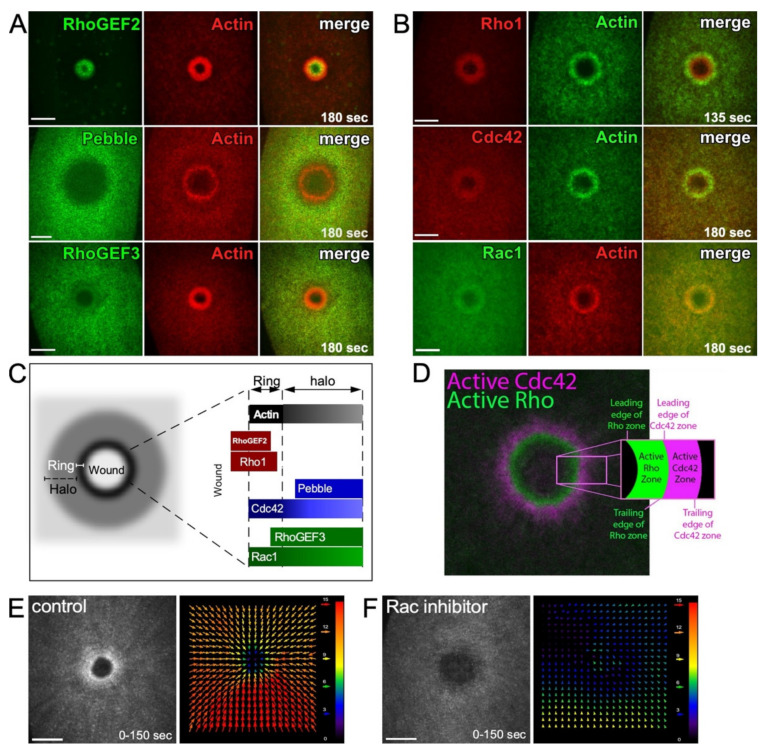
Spatial and temporal regulation of Rho family GTPases during cell wound repair. (**A**) XY projection confocal images of NC4–6 laser wounded *Drosophila* embryos co-expressing actin reporter (red) and fluorescent tagged–RhoGEFs (green; RhoGEF2, RhoGEF3, and Pebble). Used with permission from Nakamura et al., (2017), J. Cell Biol. 216(12): 3959–3969. doi: 10.1083/jcb.201704145 [27]; permission conveyed through Copyright Clearance Center, Inc. (**B**) XY projection confocal images of NC4–6 laser wounded *Drosophila* embryos co-expressing actin reporter (green in the Rho1 and Cdc42 panels and red in the Rac1 panel) and fluorescent tagged-Rho1 (red), -Cdc42 (red), or -Rac1 (green). Used with permission from Nakamura et al., (2017), J. Cell Biol. 216(12): 3959–3969. doi: 10.1083/jcb.201704145 [27]; permission conveyed through Copyright Clearance Center, Inc. (**C**) Schematic diagram showing the recruitment patterns of RhoGEFs and Rho family GTPases to wounds relative to the actin ring and halo. (**D**) XY confocal images of laser wounded *Xenopus* oocyte co-expressing Rho (green) and Cdc42 (magenta) biosensors. Reprinted from Golding et al., (2019), eLife 8: doi:10.7554/eLife.50471 [21]. (**E**,**F**) XY projection confocal images of NC4–6 laser wounded *Drosophila* embryos expressing an actin reporter in control (**E**) and Rac inhibitor injected (**F**) embryos. Cortical actin flow analysis by projection from 0 to 150 s post-wounding (left panels). Vector map of particle image velocimetry indicating actin flow speed and direction between 90 and 120 s (right panels). Arrow magnitude represents the flow rate in pixels. Scale bar: 20 µm (**A**,**B**,**E**).

**Figure 4 cells-11-02777-f004:**
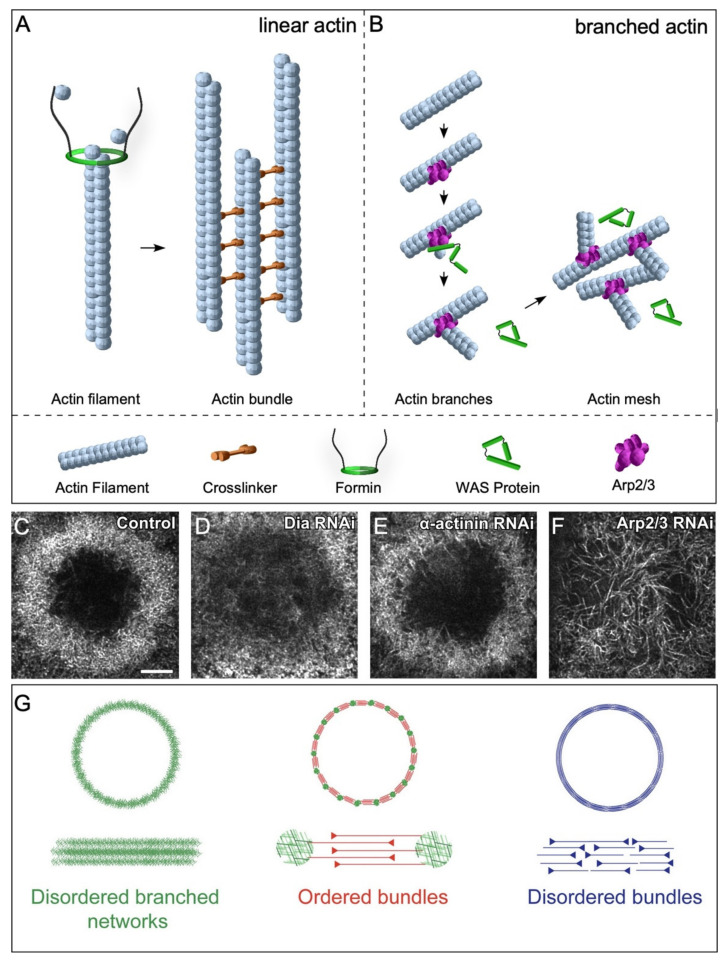
Branched and linear nucleation factors are crucial for cell wound repair. (**A**) Schematic diagram depicting crucial factors for stabilizing and nucleating linear actin filaments. Formins add actin monomers to the pointed end of the actin filament, whereas crosslinkers aid in bundling linear filaments. (**B**) Schematic diagram depicting branched filament assembly via the Arp2/3 complex and WAS family proteins. (**C**–**F**) XY super-resolution view of actin ring organization at 70% percent wound closure in control cell wounds showing a dense actin mesh circumscribing the wound (**C**), Diaphanous RNAi knockdowns (disrupting linear actin formation) exhibit a diffuse mesh of branched actin at the wound periphery (**D**), alpha-actinin RNAi knockdowns (an actin crosslinker needed for actin bundling) form a more sparse ring at the wound edge compared to that at control wounds (**E**), and Arp 2/3 RNAi knockdowns (disrupting branched actin formation) do not form an actin ring at the wound periphery, but rather have unusually long linear actin filaments within the wound (**F**). Scale bar: 5 µm. (**G**) Schematic diagram depicting different types of actin architectures. Reprinted from Ennomani et al., (2016), Curr. Biol. 26(5): 616–626. doi: 10.1016/j.cub.2015.12.069 [60], with permission from Elsevier.

**Figure 5 cells-11-02777-f005:**
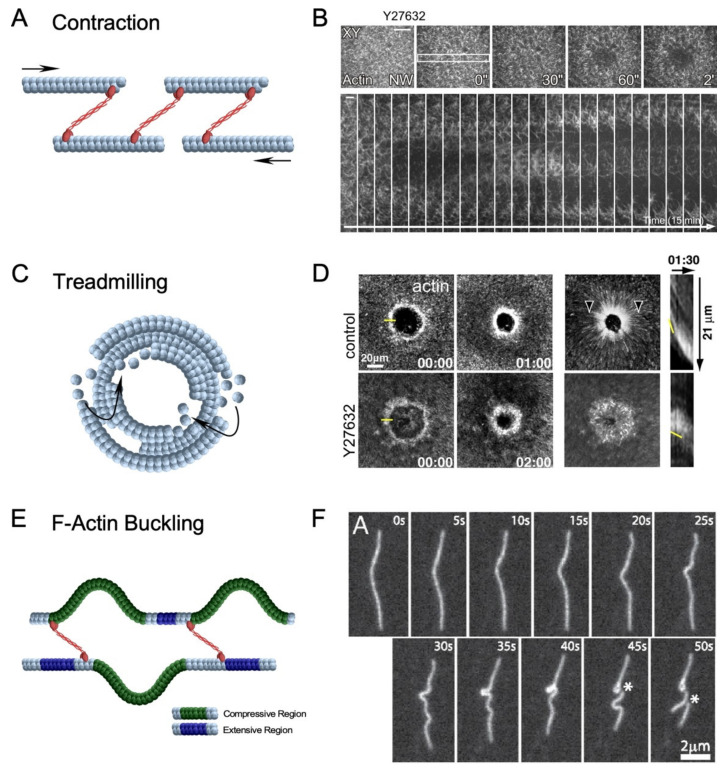
Diverse mechanisms of actin filament network contractility. (**A**) Schematic diagram depicting sarcomeric-like contraction involving non-muscle myosin II can pull actin filaments and close a wound. Arrows depict opposing movement of actin filaments. (**B**) XY and kymograph confocal micrographs of laser injured *Drosophila* syncytial embryos expressing a fluorescent actin reporter and treated with Y27632, a potent ROK inhibitor to inactivate myosin is unable to close a wound. Yellow rectangle shows region of the image used to generate the kymograph. Used with permission from Abreu Blanco et al., (2011), J. Cell Biol. 193(3): 455–464. doi: 10.1083/jcb.201011018 [9]; permission conveyed through Copyright Clearance Center, Inc. (**C**) Schematic diagram depicting actin treadmilling, which involves the continuous assembly and disassembly of actin filaments at the leading and following edges of the wound, respectively. (**D**) XY and kymograph confocal micrographs of laser injured *Xenopus* oocytes expressing an actin reporter and treated with Y27632 (myosin inhibitor). Yellow line at 00.00 indicates position in image used for kymographs, arrowheads indicate actin flow, and yellow line in kymographs shows the position of the leading edge. Reprinted from Burkel et al., (2012), Dev. Cell 23(2): 384–396. doi: 10.1016/j.devcel.2012.05.025 [25], with permission from Elsevier. (**E**) Schematic diagram depicting F-actin buckling, which involves non-uniform stress applied to a single actin filament causing it to undergo compressive (green) and extensive (purple) regions on different sections of the filament. (**F**) F-actin during contraction. Asterisks indicate position of actin filament severing. Reprinted from Murrell et al., (2012), Proc. Natl. Acad. Sci. USA 109(51): 20820–20825. doi: 10.1073.pnas [81], with permission from PNAS.

**Figure 6 cells-11-02777-f006:**
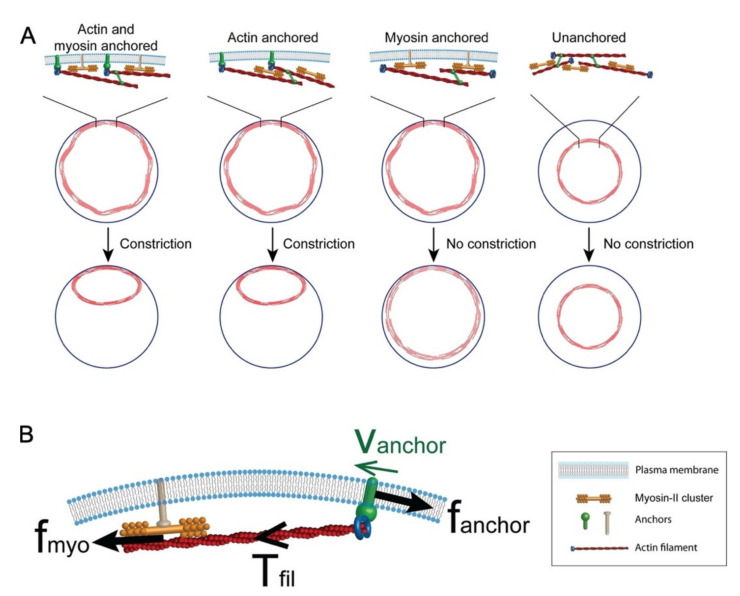
Model of cytokinetic ring attachment to the plasma membrane. (**A**) Simulation of cytokinetic rings showing that anchoring of actin to the plasma membrane is required for ring constriction. (**B**) Anchoring is important to maintain myosin-generated tension in the actin filament as depicted by the force diagram. Used with permission from Wang & O’Shaughnessy (2019), Mol. Biol. Cell 30(16): 2053–2064. doi: 10.1091/mbc.E19-03-017 [91]; permission conveyed through Copyright Clearance Center, Inc.

**Figure 7 cells-11-02777-f007:**
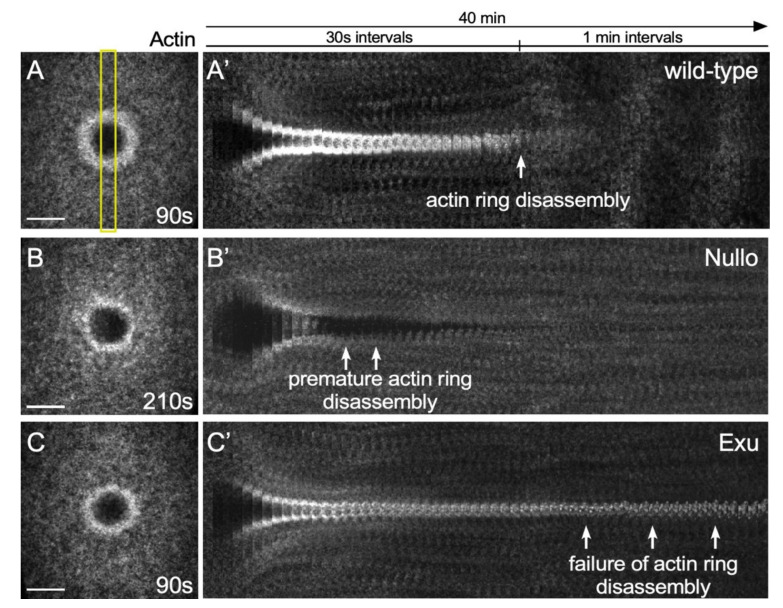
Actin remodeling phenotypes in the Drosophila cell wound model. (**A**–**C**) XY projection confocal images of NC4-6 laser wounded *Drosophila* embryos expressing an actin reporter in wild-type (**A**), Nullo RNAi (**B**), and Exu RNAi (C). Yellow rectangle shows region of the image used to generate the kymographs. (**A′**–**C′**) XY kymographs generated by cropping (yellow box) XY projection confocal images in A-C and then lining up slices for the 40 min time-lapse. Scale bar: 20 µm.

## Data Availability

Data sharing not applicable; no new datasets generated.

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
