# Peer review of "Wrangling Actin Assemblies: Actin Ring Dynamics during Cell Wound Repair"

_cells, 2022, doi:10.3390/cells11182777_

Round 1
Reviewer 1 Report
In this review, the authors describe the role of actin during cell cortex wound repair.
The review has a comprehensive summary of current literature describing the various phases of wound repair from actin ring formation, actin recruitment to wound site, and organization during wound repair.
This is a well-written and very timely review with concise summaries of the current understanding of the mechanisms by which cells have developed wound repair mechanisms and their importance in
the pathophysiologic mechanisms of diseases and malignancies. I enjoyed reading this review and only have one suggestion and one minor point:
Specific comments
In the ‘Relationship to diseases’ section, the authors give examples of diseases such as diabetes and malignancies that impair wound healing. I suggest adding the role of microbial infections such as viruses and bacterial infections to this section or a separated section ‘‘Relationship to infections’.
Minor: Figure 2 is missing label F.
Author Response
Point 1. In the ‘Relationship to diseases’ section, the authors give examples of diseases such as diabetes and malignancies that impair wound healing. I suggest adding the role of microbial infections such as viruses and bacterial infections to this section or a separated section ‘‘Relationship to infections’.
We now include a paragraph mentioning the role of microbial infections in cell wound repair and have amended the title of this section to ‘Relationship to diseases and infections’.
Point 2. Figure 2 is missing label F.
(now Figure 3) We have added the missing F.
Reviewer 2 Report
This is a comprehensive review article focusing on actin dynamics during plasma membrane wound repair. The relevant literature with respect to actin remodeling during this cellular stress response is covered and the manuscript is well-written and illustrated.
However, I have one major concern with respect to the generality of the findings discussed. The authors focus their review on mechanisms involved in the formation of an actin ring around the injured plasma membrane and already in figure 1 show a (postulated) mechanism of single cell wound repair that critically relies on the formation of this actin ring and its actomyosin-driven purse-string constriction that pulls the wound closed for repair. Moreover, this mechanism involves the formation of a membrane patch at the wound site responsible for immediate plugging of the injury site. However, to the knowledge of this reviewer such mechanism has only been shown operate in exceptionally large cells such as Xenopus eggs and Drosophila embryos (the latter mainly by the authors’ group). For many other cell types, in particular somatic tissue cells, and other wounding scenarios, e.g. pore forming toxins, formation of membrane patches and a purse-string actomyosin-dependent closing of the wound that precedes the actual membrane resealing have not been reported and/or identified. To name just a few of many such examples (no vesicular membrane patch, no actomyosin ring): membrane resealing in MCF7 cancer cells (Jaiswal et al., Nat Commun 2014; Sønder et al., Sci Adv 2021), muscle cells (Marg et al., Traffic 2012; Demonbreun et al., J Cell Biol 2016), endothelial cells (Ashraf and Gerke, BBA 2021), Dictyostelium (Yumura Biol Open 2014). Actin rearrangements are also observed in these examples but they happen after the actual membrane wound has been resealed and they don’t occur in a purse-string type arrangement.
Thus, either the authors subject their manuscript to an extensive revision to also include other mechanisms of membrane repair at the single cell level (e.g. resealing by endocytosis, exocytosis, plasma membrane vesicle shedding), which all seem to happen independently of actin (in the actual resealing phase), or they restrict their review to membrane and in particular actin rearrangements in the course of wound repair in exceptionally large cells. I think the latter is fine but this would have to mentioned in the title and throughout the text.
Other points that should be addressed.
1. Throughout the text the authors refer to two of their studies which have been deposited in BioRxiv. I think only peer-reviewed papers should be cited, the non-published data could however be included if shown.
2. Fig 1. What actually happens to the temporary membrane patch. Perhaps the authors could add a sentence or two here.
3. Fig 3. Cartoon is based on unpublished data (those of the BioRxiv manuscript). Results revealing the involvement of Spire should be shown (see above). Results shown in C-F need a more detailed description in the legend.
4. Lines 110-127 and 218-222. The time course of actin ring formation and translocation towards the wound in comparison to the actual membrane closure (resealing) should be discussed. Does membrane resealing require actin ring translocation, i.e. occur after the translocation. The authors say that in Drosophila the actin ring translocation requires 10 min or more. Does this mean the actual membrane wound stays open for 10 or even more min?
5. Lines 135-137. Misregulation of actin networks results in infertility ... Of course correct. But it this related to wound repair? If so, explain; if not take out.
6. Lines 322-324. Annexins are no transmembrane proteins.
7. Chapter 5 – Relationship to disease. Here the muscle diseases that had been linked early-on to defects in membrane repair should be mentioned, e.g. Limb Girdle muscular dystrophy and the dysferlin gene.
8. Line 458. Why is the Ca influx uniform? It only occurs at the wound site, thus is spatially restricted.
Author Response
- Thus, either the authors subject their manuscript to an extensive revision to also include other mechanisms of membrane repair at the single cell level (e.g. resealing by endocytosis, exocytosis, plasma membrane vesicle shedding), which all seem to happen independently of actin (in the actual resealing phase), or they restrict their review to membrane and in particular actin rearrangements in the course of wound repair in exceptionally large cells. I think the latter is fine but this would have to mentioned in the title and throughout the text.
We take the point of the Reviewer and have amended the text and title to make it clear that our focus is on actin ring dynamics in the larger cell wound models in this review.
- Throughout the text the authors refer to two of their studies which have been deposited in BioRxiv. I think only peer-reviewed papers should be cited, the non-published data could however be included if shown.
One of these BioRxiv studies has been published and the reference has been updated. The other study is in revision and will be published soon. Cells allows the citation of BioRxiv deposited papers. Since the remaining BioRxiv study is cited as an example, rather than as proof of a given point, we still include this citation.
- Fig 1. What actually happens to the temporary membrane patch. Perhaps the authors could add a sentence or two here.
The temporary membrane patch is eventually removed from the wound site. We have added a sentence to the text and figure legend to state this.
We now include XY and XZ projection images, as well as a dynamic kymograph, showing both actin (fluorescent actin reporter) and membrane (FM4-64) dynamics in the Drosophila cell wound model (new Figure 2).
- Fig 3. Cartoon is based on unpublished data (those of the BioRxiv manuscript). Results revealing the involvement of Spire should be shown (see above). Results shown in C-F need a more detailed description in the legend.
(now Figure 4) The cartoons in panels A-B are based on numerous studies in the field and were only intended to show the basics of linear and branched actin formation and their subsequent assembly into bundles or meshes, respectively. We had also planned to focus on formins in this review. While Spire is well-known as a de novo linear actin nucleation factor (hence its inclusion in the graphic), we take the point of the reviewer that its role in wound repair is not yet published. We have amended the figure to remove Spire and simplified the graphics depicting linear and branched actin nucleation.
We now include a more detailed description of the images shown in panels C-F in the figure legend.
- Lines 110-127 and 218-222. The time course of actin ring formation and translocation towards the wound in comparison to the actual membrane closure (resealing) should be discussed. Does membrane resealing require actin ring translocation, i.e. occur after the translocation. The authors say that in Drosophila the actin ring translocation requires 10 min or more. Does this mean the actual membrane wound stays open for 10 or even more min?
We previously published a paper showing that the translocation of the actin ring is highly linked to that of the membrane at the wound site during cell wound repair in the Drosophila cell wound model (PMID: 21518790). We now include actin and membrane dynamics during cell wound repair in the Drosophila model in Figure 2 and have amended the text to mention actin ring and membrane dynamics in the Introduction. Section 3.2 also includes a discussion on the translocation of an actin ring and plasma membrane.
- Lines 135-137. Misregulation of actin networks results in infertility ... Of course correct. But it this related to wound repair? If so, explain; if not take out.
This sentence was included for context, but we take the reviewer’s point and have removed the sentence.
- Lines 322-324. Annexins are no transmembrane proteins.
This sentence has been amended.
Reviewer 3 Report
The manuscript by Hui et al is about an overview of actin cytoskeleton dynamics in cell wound repair. This important area has been studied more than 20 years, and a great deal of knowledge has been accumulated. The authors have challenged a comprehensive review of actin dynamics in wound repair, however, they fail in this submitted manuscript. This manuscript is messy and unclear overall, and there are some obvious errors. I do not believe that the authors fully understand actin dynamics. I even doubt that the corresponding author has read the submitted manuscript properly. In order not to spread the wrong perception, I do not agree with the publication of this manuscript. At least, the authors should focus on the topic(s) of their expertise, and rewrite carefully.
There are many problems with this article, but I point out three that I find particularly important. Again, there are too many problems with this manuscript to list.
1. Fig. 4C, I have been working on actin dynamics for over 20 years, and have never seen an actin treadmilling model like this. Do the references suggest such a model? Or, is this a figment of the authors’ imagination?
2. Page 9, 3.1.3. F-actin buckling model. It seems there is no studies reporting F-actin buckling is involved in wound repair.
3. Fig. 3A legend. The authors mention as “formins add actin monomers to the pointed end.” This is completely wrong. In addition, why is Spire that is not mentioned in the text shown in the figure?
Author Response
The manuscript by Hui et al is about an overview of actin cytoskeleton dynamics in cell wound repair. This important area has been studied more than 20 years, and a great deal of knowledge has been accumulated. The authors have challenged a comprehensive review of actin dynamics in wound repair, however, they fail in this submitted manuscript. This manuscript is messy and unclear overall, and there are some obvious errors. I do not believe that the authors fully understand actin dynamics. I even doubt that the corresponding author has read the submitted manuscript properly. In order not to spread the wrong perception, I do not agree with the publication of this manuscript. At least, the authors should focus on the topic(s) of their expertise, and rewrite carefully.
There are many problems with this article, but I point out three that I find particularly important. Again, there are too many problems with this manuscript to list.
- Fig. 4C, I have been working on actin dynamics for over 20 years, and have never seen an actin treadmilling model like this. Do the references suggest such a model? Or, is this a figment of the authors’ imagination?
We have amended our graphic of actin treadmilling in cell wound repair to depict more clearly G-actin monomer incorporation at the leading edge and F-actin disassembly into G-actin monomers at the trailing edge of the actin ring.
- Page 9, 3.1.3. F-actin buckling model. It seems there is no studies reporting F-actin buckling is involved in wound repair.
Although F-actin buckling has not yet been demonstrated in a cell wound repair context, this is a major model for actin ring translocation and can play a vital role on the efficiency of actin ring contraction as described by Ennomani et al. (2016). Therefore, we feel that the reader will benefit from its inclusion in this section describing F-actin buckling as a mechanism for actin filament contractility. We have amended the title for Figure 4 (now Figure 5) to make this distinction clearer.
- Fig. 3A legend. The authors mention as “formins add actin monomers to the pointed end.” This is completely wrong. In addition, why is Spire that is not mentioned in the text shown in the figure?
We have corrected the legend to Figure 3A (now Figure 4A).
As mentioned in point 3 for Reviewer 2, while Spire is well-known as a de novo linear actin nucleation factor (hence its inclusion in the graphic), we take the point of the reviewer that its role in cell wound repair is not yet published. We have amended the figure to remove Spire.
Round 2
Reviewer 2 Report
In this revised version the authors address my points of concern. However, while some of my criticisms have been dealt with appropriately, other aspects are still problematic. In particular, my major concern has not been addressed fully. This related to the generality of the findings discussed. After reading the review in its present form the non-expert reader arrives at the perception that every plasma membrane wound is repaired by means of a membrane patch forming at the wound site and an actin ring that circles the wound and then translocates in the direction of the wound. However, as pointed out in my initial review this has only been shown for exceptionally large cells, eg Xenopus oocytes and Drosophila embryos. Other cells (somatic cells, tumor cells and others) show membrane rearrangements and certain actin dynamics but in these cases a membrane patch or plug and a concentric actin ring have never been observed. Therefore I concluded in my review that the authors should either discuss other mechanisms of membrane repair at the single cell level (e.g. resealing by endocytosis, exocytosis, plasma membrane vesicle shedding), which all seem to happen independently of actin (in the actual resealing phase), or they should restrict their review to membrane and in particular actin rearrangements in the course of wound repair in exceptionally large cells. The latter would mean that this would have to be mentioned in the title and throughout the text.
The authors responded to this point by changing the title from ‘Wrangling actin assemblies: cytoskeleton dynamics in cell wound repair‘ to ‘Wrangling actin assemblies: actin ring dynamics in cell wound repair‘ and included a few mentionings of the large cells in the Introduction and first parts of the review. However, this is not sufficient because the title does not tell the reader that only aspects of membrane repair in these large cells are discussed. Something like ‘Wrangling actin assemblies: actin ring dynamics in wound repair in very large cells‘ would be appropriate. Moreover, it has to be mentioned explicitedly in the abstract that only these type of wounds/cells are discussed. Therefore, the second sentence of the abstract should be changed to something like ‘In large cells such as oocytes, it is evident that once a wound is recognized, the cell must rapidly plug ...‘. And the third sentence should end ‘...during the repair process in very large cells such as oocytes‘. The next sentence should then start ‘In these cells, proteins such as ...‘. Finally the last sentence should also contain this restriction ‘...of the actin ring during the cell wound repair process in exceptionally large cells, as well as the overall ...‘. In addition to these amendments in the abstract, the title of figure 1 should be changed to ‘Aspects of wound repair in exceptionally large cells‘.
A second point that was not appropriately addressed concerns the time course of membrane resealing and actin ring translocation. In my initial point 4, I suggested that the time course of actin ring formation and translocation towards the wound should be discussed in comparison to the actual membrane closure, the resealing. The authors responded to this point by showing images of membrane stain (FM dye) in comparison to actin ring formation in the Drosophila model (the new figure 2). This is informative but it doesn’t answer my question. I was wondering when the plasma membrane is actually closed, ie resealed again. The figure only shows that membrane material accumulates at the wound (presumably the membrane plug) but it doesn’t reveal whether the plasma membrane hole is still open or resealed, for instance can material such as Ca2+ ions still pass through the wound? This does not have to be addressed experimentally (it’s a review after all) but it should be discussed.
The authors might have missed two of my earlier points. Below I just copy/pasted them from my initial review:
7. Chapter 5 – Relationship to disease. Here the muscle diseases that had been linked early-on to defects in membrane repair should be mentioned, e.g. Limb Girdle muscular dystrophy and the dysferlin gene.
8. Line 458. Why is the Ca influx uniform? It only occurs at the wound site, thus is spatially restricted.
Author Response
In this revised version the authors address my points of concern. However, while some of my criticisms have been dealt with appropriately, other aspects are still problematic. In particular, my major concern has not been addressed fully. This related to the generality of the findings discussed. After reading the review in its present form the non-expert reader arrives at the perception that every plasma membrane wound is repaired by means of a membrane patch forming at the wound site and an actin ring that circles the wound and then translocates in the direction of the wound. However, as pointed out in my initial review this has only been shown for exceptionally large cells, eg Xenopus oocytes and Drosophila embryos. Other cells (somatic cells, tumor cells and others) show membrane rearrangements and certain actin dynamics but in these cases a membrane patch or plug and a concentric actin ring have never been observed. Therefore I concluded in my review that the authors should either discuss other mechanisms of membrane repair at the single cell level (e.g. resealing by endocytosis, exocytosis, plasma membrane vesicle shedding), which all seem to happen independently of actin (in the actual resealing phase), or they should restrict their review to membrane and in particular actin rearrangements in the course of wound repair in exceptionally large cells. The latter would mean that this would have to be mentioned in the title and throughout the text.
The authors responded to this point by changing the title from ‘Wrangling actin assemblies: cytoskeleton dynamics in cell wound repair‘ to ‘Wrangling actin assemblies: actin ring dynamics in cell wound repair‘ and included a few mentionings of the large cells in the Introduction and first parts of the review. However, this is not sufficient because the title does not tell the reader that only aspects of membrane repair in these large cells are discussed. Something like ‘Wrangling actin assemblies: actin ring dynamics in wound repair in very large cells‘ would be appropriate. Moreover, it has to be mentioned explicitedly in the abstract that only these type of wounds/cells are discussed. Therefore, the second sentence of the abstract should be changed to something like ‘In large cells such as oocytes, it is evident that once a wound is recognized, the cell must rapidly plug ...‘. And the third sentence should end ‘...during the repair process in very large cells such as oocytes‘. The next sentence should then start ‘In these cells, proteins such as ...‘. Finally the last sentence should also contain this restriction ‘...of the actin ring during the cell wound repair process in exceptionally large cells, as well as the overall ...‘. In addition to these amendments in the abstract, the title of figure 1 should be changed to ‘Aspects of wound repair in exceptionally large cells‘.
While we understand the point that the reviewer is making, we do not fully agree with their viewpoint. The model organism models used for cell wound repair (Xenopus oocytes, Drosophila syncytial embryos) tend to be larger cells, however, the actin ring dynamics that we are discussing in this review do happen in smaller cells. The Bement lab previously showed that the same actomyosin ring and concentric rings of Rho GTPases as seen in Xenopus oocyte and Drosophila embryo models are also observed in Xenopus epithelial cells during cell wound repair (PMID: 19631537 and 31647414). The size of Xenopus epithelial cells is approximately 20 µm in diameter, which is very similar to tissue culture cells used in the cell wound repair studies (e.g. 17µm diameter for HUVEC cells, 20-25µm diameter for MCF7 cells, and 20-40 µm diameter for HeLa cells depending on culture conditions, PMID: 34324830 and 33667528). We now include an example of the Xenopus embryo where a single cell was wounded within the context of the epithelia (Figure 1E). We make it clear that the precise mechanisms of repair are context-dependent and that this particular review is focusing on actin ring dynamics in cell wound repair models that use an actin ring.
A second point that was not appropriately addressed concerns the time course of membrane resealing and actin ring translocation. In my initial point 4, I suggested that the time course of actin ring formation and translocation towards the wound should be discussed in comparison to the actual membrane closure, the resealing. The authors responded to this point by showing images of membrane stain (FM dye) in comparison to actin ring formation in the Drosophila model (the new figure 2). This is informative but it doesn’t answer my question. I was wondering when the plasma membrane is actually closed, ie resealed again. The figure only shows that membrane material accumulates at the wound (presumably the membrane plug) but it doesn’t reveal whether the plasma membrane hole is still open or resealed, for instance can material such as Ca2+ ions still pass through the wound? This does not have to be addressed experimentally (it’s a review after all) but it should be discussed.
We are not sure what the Reviewer actually means about resealing. In our model as shown in Figure 1, membrane/vesicles are recruited to the wound and then form a temporary membrane patch to plug the hole. This membrane patch attaches to the plasma membrane at the wound periphery and forms a seal such that calcium can no longer enter the cell through the wound after 1 min (as shown in Figure 1B and the movie published in PMID: 33306674). After the hole is sealed, an actomyosin ring forms that is attached to the overlying plasma membrane through at least E-Cadherin – which pulls the plasma membrane closed as the actomyosin ring contracts. The vesicle mediated membrane patch is pushed inward to the cell and eventually removed from the wound as the closing plasma membrane constricts. These processes are based on previous studies using GCaMP, FM4-64 dye, several membrane markers, and E-Cadherin-GFP and are also described in several reviews (e.g. PMID: 34639226, 28596334, and 26336031). As the focus of this review is actin ring dynamics, an extensive discussion about resealing the hole by membrane/vesicle dynamics is beyond the scope of the review.
The authors might have missed two of my earlier points. Below I just copy/pasted them from my initial review:
- Chapter 5 – Relationship to disease. Here the muscle diseases that had been linked early-on to defects in membrane repair should be mentioned, e.g. Limb Girdle muscular dystrophy and the dysferlin gene.
We are sorry that we missed these points in the previous revision. We now include a paragraph mentioning muscle diseases in the Relationship to diseases and infections section.
- Line 458. Why is the Ca influx uniform? It only occurs at the wound site, thus is spatially restricted.
We had intended to mean uniform calcium diffusion from the wound site, but take the point of how this phrasing can be interpreted differently. We have removed "uniform" from the text.
Reviewer 3 Report
In the revised manuscript, the schematic diagrams in the new Fig. 5 (previous Fig. 4) are still not good enough. In Fig. 5C (Treadmilling), the force of actin assembly should push plasma membrane to close a wound. However, in Fig. 5C actin filaments are arranged in parallel with the plasma membrane (that is not shown though). Please take a moment to consider whether actin polymerization at the plus-ends can push the membrane in such a situation. Than would not be possible.
The schemes of Fig. 5A and E also do not make sense. The situations depicted in Fig. 5A (Contraction) and E (F-actin Buckling) are basically same. If actin filaments are long and cross-linked at two (or more) points with myosin, does buckling occur rather than contraction? I do not think so. For example, if the authors see the new Fig. 6 (which is taken from a reference), the authors will understand that the schematic diagrams in Fig. 5 are poor. Before publishing the manuscript. I suggest to improve the Fig. 5A, C, E, or simply remove them.
Author Response
In the revised manuscript, the schematic diagrams in the new Fig. 5 (previous Fig. 4) are still not good enough. In Fig. 5C (Treadmilling), the force of actin assembly should push plasma membrane to close a wound. However, in Fig. 5C actin filaments are arranged in parallel with the plasma membrane (that is not shown though). Please take a moment to consider whether actin polymerization at the plus-ends can push the membrane in such a situation. Than would not be possible.
It is our intention to depict the dynamics of actin filaments under various constriction mechanisms to generate contractility in actin networks. Thus, our focus is on actin filaments rather than the plasma membrane.
Schematics of actin ring treadmilling have been shown in this way in previous publications (PMID: 34639226 and 30282661). In the Drosophila model, the actomyosin ring is attached to the overlying plasma membrane through at least E-Cadherin – which facilitates pulling the plasma membrane closed as the actomyosin ring contracts (PMID: 21518790). Therefore, as actin is polymerized at the leading edge of the wound, E-Cadherin is linking the actin ring and the plasma membrane to pull the wound closed.
The schemes of Fig. 5A and E also do not make sense. The situations depicted in Fig. 5A (Contraction) and E (F-actin Buckling) are basically same. If actin filaments are long and cross-linked at two (or more) points with myosin, does buckling occur rather than contraction? I do not think so. For example, if the authors see the new Fig. 6 (which is taken from a reference), the authors will understand that the schematic diagrams in Fig. 5 are poor. Before publishing the manuscript. I suggest to improve the Fig. 5A, C, E, or simply remove them.
We respectfully disagree with the reviewer and feel that our schematics are adequately depicting the various actin filament constriction mechanisms. Actin filament contraction and buckling have been similarly depicted in this way and discussed to be different in previous publications (PMID: 27505246, 29482169, 23213249, 23003998, 24382887). Previous publications have proposed F-actin buckling as a result of non-uniform myosin distribution and/or non-uniform myosin velocities on actin filaments (PMID: 27505246 and 24382887). Therefore, it is possible that actin filaments linked at two or more points with myosin will exhibit filament buckling. It is not our intention to suggest that F-actin contraction and buckling are mutually exclusive events. Although the exact connectivity and architecture of the actin network can dictate which mechanism is dominant, network contractility is likely best described using a combination these mechanisms (PMID: 26898468).